

# Thermobarokinetics of ice: constitutive formulation for the coupled effect of temperature, stress, and strain rate in ice

Faranak Sahragard, Mehdi Pouragha, and Mohammad Rayhani

Department of Civil and Environmental Engineering, Carleton University, 1125 Colonel By Dr, K1S5B6, Ottawa, Ontario, Canada

**Correspondence:** Mehdi Pouragha (mehdi.pouragha@carleton.ca)

**Abstract.** Understanding and modeling the mechanical behavior of ice under varying thermal and loading conditions is essential for cryospheric science, permafrost engineering, and the design of polar infrastructure. A central challenge lies in capturing the strong coupling between stress, strain rate, and temperature, an interdependence referred to in this work as the thermobarokinetics of ice. This study presents a three-dimensional constitutive model that explicitly incorporates this coupling through a unified thermomechanical framework. Notably, the model employs shared functional dependencies for both viscosity and damage initiation, allowing key rate- and temperature-sensitive processes to be represented using a minimal set of physically interpretable parameters. Damage evolution is governed by an energy-based law that depends on strain rate and temperature. The model is calibrated and validated against triaxial compression and relaxation test data on polycrystalline ice, demonstrating its ability to capture salient features of ice mechanics such as ductile to brittle transitions, strain-rate-dependent strength, stress relaxation, and thermal softening. In addition, a novel healing mechanism inspired by viscous sintering is introduced, in which the rate of damage reversal is driven by viscous energy dissipation and modulated by pressure and temperature.

## 1 Introduction

Accurate modeling of ice mechanics is essential in a wide range of disciplines, including glaciology, cryospheric science, geotechnical engineering of cold regions, and permafrost engineering. However, developing robust constitutive models for ice remains a challenge, primarily due to the multitude of relevant physical processes and parameters, and the complex, often nonlinear interactions among them. In particular, the strong coupling between pressure, strain rate, and temperature effects makes it exceedingly difficult to isolate individual mechanisms for separate analysis and modeling—as is more commonly feasible for other natural solids such as metals or soils. For instance, careful experimental studies have shown that strain rate dependency in ice not only induces creep and relaxation behavior, but also affect the yield stress and can influence the brittle or ductile nature of failure (Jones, 1982; Jones and Chew, 1983; Durham et al., 1983; Schulson, 1990; Murrell et al., 1991; Kalifa et al., 1992; Rist and Murrell, 1994; Gagnon and Gammon, 1995; Gratz and Schulson, 1997; Sammonds et al., 1998; Mizuno, 1998; Meglis et al., 1999). Similarly, pressure influences not only the yield stress of ice but also its viscous behavior, while temperature affects nearly all mechanical properties, including stiffness, yield strength, and viscosity. The intertwined effects of temperature, pressure and strain rate introduce strong nonlinearities into the material response that cannot be captured by



simple additive formulations. Consequently, the reliable interpretation of experimental data and in-situ observations demands
a theoretical framework that accounts, at least partially, for the coupled influence of these parameters. For example, in geotech-
nical engineering, it is not uncommon to interpret the failure of frozen soils or ice using cohesive-frictional Mohr-Coulomb
criteria, without adequately considering the underlying viscous mechanisms. However, the friction and cohesion values de-
rived in this way are inherently limited in accuracy and are unlikely to remain valid when extrapolated beyond the specific
experimental conditions under which they were obtained.

By nature, ice exhibits behaviors characteristic of fluids, granular materials, and crystalline solids. It is therefore not sur-
prising that constitutive models developed in these fields have been adapted and extended to describe the thermomechanical
response of ice. Some instances include: ice models based on rheological models for the viscous flow (e.g., Nye, 1953; Glen,
1955; Nye, 1957; Azuma, 1994); models based on granular flow (e.g., Wilchinsky and Feltham, 2006; Herman, 2022; Ren
et al., 2025); models based on granular solids (e.g., Balendran and Nemat-Nasser, 1993; Tremblay and Mysak, 1997); and
extensions to the crystal mechanics models (e.g., Michel, 1978; Cole, 1995, 1998, 2020).

Combining various physics such as viscosity, elasticity, and plasticity is commonly achieved through using rheological
models built from various configurations of springs, dashpots, and frictional elements. One common approach is the so-called
Burgers body model, which consists of a Maxwell element in series with a Kelvin element. In this model, total strain is
decomposed into instantaneous elastic, delayed elastic, and permanent viscous components. One of the earliest applications
of such a viscoelastic model to ice is found in the work of Sinha (1978), who developed a one-dimensional formulation to
capture the creep behavior of ice under constant stress. The delayed elastic response was expressed as a function of both time
and temperature, while Glen-Nye flow law (Nye, 1953; Glen, 1955) was employed to describe the viscous strain. Subsequent
developments extended the model to three-dimensional stress and strain states and incorporated damage mechanisms. For
example, Karr and Choi (1989) introduced a damage parameter to capture anisotropic microcrack evolution, accounting for its
dependence on stress and strain rate. Later, Duddu and Waisman (2012) elaborated on this model by incorporating temperature-
dependent viscous strain, though their formulation remains limited to low stress levels and strain rates. Another notable three-
dimensional version of the Burgers model was developed by Xiao and Jordaan (1996), who employed nonlinear dashpots
following a power law in both the Kelvin and Maxwell elements. Their approach integrated Schapery's viscoelastic damage
theory (Schapery, 1991) to represent creep-induced damage and accounted for the suppressive effect of confining pressure
on microcrack growth. However, this model did not explicitly include temperature effects. More recently, Xu et al. (2019)
incorporated plasticity into the Burgers model. Their formulation included temperature-dependent strain rates for both delayed
elastic and viscous components. The influence of confining pressure was also captured: strain rate was found to decrease with
increasing pressure up to $50\,\mathrm{MPa}$, beyond which it increased. Nevertheless, this model remains limited to ductile deformation
and does not address brittle damage mechanisms.

Developing multi-mechanism constitutive models involves a careful balance between the complexity of the overall rheo-
logical configuration and the sophistication of the individual constitutive formulations that represent each mechanism. This
balance becomes especially important when accounting for the coupling between mechanisms, which is typically embedded
within the constitutive description of each component. Indeed, in some cases, simpler rheological elements, such as Maxwell



units, are also successfully employed to model the behavior of ice. One prominent such example is found in the works of
Dansereau and coworkers (Dansereau et al., 2016, 2017; Olason et al., 2022) who introduced a Maxwell elasto-brittle model
for sea ice that incorporates the evolution of both the elastic modulus and the viscosity, while simultaneously accounting for
damage and healing mechanisms. Despite its relative simplicity, the model offers unique advantages due to its minimal number
of parameters, straightforward configuration, and the incorporation of a healing effect essential for accurately simulating ice

behavior under long-term loading and high pressure conditions (Brodeau et al., 2024). Some of the simplifications particular
to sea ice adopted in this model include temperature independence, a Mohr-Coulomb damage initiation function excluding
damage under isotropic stresses, and time-driven damage and healing evolution laws.

A review of the existing literature on constitutive modeling of ice points to the persistent challenge of accurately formulating
the coupling between stress, strain rate, and temperature effects remains a challenging task. While the general frameworks of

rational mechanics for these couplings is laid out in monographs such as Hutter (2020), in practice, the existing constitutive
models for ice often struggle to coherently integrate these interactions. As a result, their applicability is typically confined to a
narrow range of conditions, such as specific temperatures or loading regimes, thereby limiting their generalizability across the
wide spectrum of scenarios encountered in cryospheric and engineering applications.

In this study, we aim to develop a three-dimensional constitutive model for ice that addresses key shortcomings of existing

approaches and introduces an integrated formulation to capture the combined effects of temperature, pressure, and strain
rate; referred to here as the *thermobarokinetics* of ice. The model builds on the Maxwell elasto-brittle framework developed
by Dansereau et al. (2016), with several key improvements including stress and temperature dependent viscosity, energy-
driven damage, and temperature and rate dependent damage initiation limit, while keeping the model parameters manageable.
A deliberate effort was made to integrate existing constitutive formulations from the literature into a coherent framework,

rather than introducing entirely new modeling components. We adopt the Glen–Nye flow law (Nye, 1953; Glen, 1955) as
a non-Newtonian viscosity model, combined with an Arrhenius-type expression to capture the temperature dependence of
both viscosity and the energy threshold for damage initiation. Comparisons with available experimental data on ice support
the plausibility of the proposed constitutive model in reproducing key features of ice behavior across a range of pressures,
temperatures, and strain rates. Additionally, we propose a formulation for the healing process, drawing inspiration from viscous

sintering phenomena observed in materials such as glass and magma.

## 2  Mathematical notation

Throughout this article, we use bold characters (e.g. $\mathbf{I}$ and $\varepsilon$) for second-order tensors, and hollow characters (e.g. $\mathbb{C}$) for
fourth-order tensors. The colon symbol : represents a double-contraction, i.e. $\mathbf{A} : \mathbf{B} = A_{ij}B_{ij}$ and $\mathbb{C} : \mathbf{A} = C_{ijkl}A_{kl}$, while the
symbol $\otimes$ represents dyadic product, i.e. $\mathbb{C} = \mathbf{A} \otimes \mathbf{B}$ is equivalent to $C_{ijkl} = A_{ij}B_{kl}$. The second-order tensor of infinitesimal



strain, $\varepsilon$, and the Cauchy stress $\boldsymbol{\sigma}$ can be divided into their spherical and deviatoric parts:

$$\varepsilon_{vol} = \boldsymbol{\varepsilon} : \mathbf{I}, \qquad \boldsymbol{e} = \boldsymbol{\varepsilon} - \frac{\varepsilon_{vol}}{3}\mathbf{I}, \qquad \varepsilon_q = \sqrt{\frac{2}{3}\boldsymbol{e} : \boldsymbol{e}}$$

$$p = \frac{1}{3}\boldsymbol{\sigma} : \mathbf{I}, \qquad \boldsymbol{s} = \boldsymbol{\varepsilon} - p\mathbf{I}, \qquad q = \sqrt{\frac{3}{2}\boldsymbol{s} : \boldsymbol{s}} \tag{1}$$

where $\varepsilon_{vol}$ is the volumetric strain, $\boldsymbol{e}$ is the deviatoric strain tensor, $\varepsilon_q$ is the deviatoric strain, $p$ is the mean stress, $\boldsymbol{s}$ is deviatoric stress tensor, $q$ is the deviatoric stress, and $\mathbf{I}$ is the second-order identity tensor. A fourth-order identity tensor, $\mathbb{I}$, is also defined as $\mathbb{I}_{ijkl}^{(4)} = \frac{1}{2}\left(\delta_{ik}\delta_{jl} + \delta_{il}\delta_{jk}\right)$ with $\delta$ being the Kronecker delta. Finally, the over-dot $\dot{x} = \frac{\partial x}{\partial t}$ represents the time-rate.

## 3  Model formulation

The sea-ice model by Dansereau et al. (2016), used as a starting point herein, adopts a viscoelastic Maxwell configuration, as shown in Fig. 1, including also a damage mechanism. The damage initiation is described by a non-evolving Mohr-Coulomb criterion in terms of the undamaged stresses, while the stress, elastic stiffness, and the Newtonian viscosity are prescribed to degrade with damage. Temporal evolution of the damage is explicitly given in the model, with damage reversal, or healing, also made possible through a constant healing rate.

The model developed in this study also adopt a similar viscoelastic Maxwell configuration which implies that the two mechanisms (i.e. viscous and elastic) share a stress while the strains are added:

$$\boldsymbol{\varepsilon} = \boldsymbol{\varepsilon}^e + \boldsymbol{\varepsilon}^v \tag{2}$$

with the superscripts $e$ and $v$ representing elastic and viscous mechanisms. The viscosity formulation presented later will indicate that the viscous strain, $\boldsymbol{\varepsilon}^v$ is traceless and deviatoric, implying that the volumetric strain is only due to the elastic component, i. e. $\varepsilon_{vol} = \varepsilon_{vol}^e$.

Using the effective stress concept from damage mechanics and the principle of strain equivalence, the elastic energy density, $\psi$, of the elastic component under isothermal conditions is expressed as (Lemaitre and Chaboche, 1994; Chaboche, 1987):

$$\psi(\boldsymbol{\varepsilon}^e, D) = \frac{1}{2}(1-D)\boldsymbol{\varepsilon}^e : \mathbb{C}^e : \boldsymbol{\varepsilon}^e \tag{3}$$

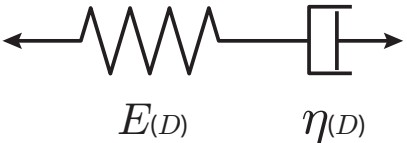

**Figure 1.** Schematic of the one-dimensional Maxwell model for a continuum material with elastic modulus $E$ and viscosity $\eta$. Both elastic modulus and viscosity depend on the damage level, $D$.



where $\mathbb{C}^e$ is the elasticity tensor for undamaged material, and $D \in [0, 1]$ describes the damage level. The inclusion of the $1 - D$ indicates the release of elastic energy as a result of damage accumulation. Elastic Cauchy stress is defined as the conjugate to the elastic strain:

$$\boldsymbol{\sigma} = \frac{\partial \psi(\boldsymbol{\varepsilon}^e, D)}{\partial \boldsymbol{\varepsilon}^e} = (1 - D)\mathbb{C}^e : \boldsymbol{\varepsilon}^e = (1 - D)\boldsymbol{\sigma}^* \tag{4}$$

where $\boldsymbol{\sigma}^*$ is the undamaged stress tensor. A thermodynamic force, $Y$, is also defined as the conjugate to the damage parameter $D$, which represents the damage driving force within the material:

$$Y = -\frac{\partial \psi(\boldsymbol{\varepsilon}^e, D)}{\partial D} = \frac{1}{2}\boldsymbol{\varepsilon}^e : \mathbb{C}^e : \boldsymbol{\varepsilon}^e \tag{5}$$

The thermodynamic force, $Y$, corresponds to the rate of elastic strain energy released per unit increase in damage within the material (Chaboche, 1987), which is equivalent to the elastic strain energy in the undamaged state of the material. An isotropic elasticity is adopted in this study, with the elastic modulus given as:

$$\mathbb{C}^e = \frac{E}{3(1 - 2\nu)} \mathbf{I} \otimes \mathbf{I} + \frac{E}{(1 + \nu)} \left( \mathbb{I} - \frac{1}{3}\mathbf{I} \otimes \mathbf{I} \right) \tag{6}$$

with $E$ being elastic Young's modulus of the undamaged material, and $\nu$ being the Poisson's ratio. The literature on the potential effects of temperature on the elastic stiffness of ice remains inconclusive, with experimental interpretations often influenced by the specific constitutive models employed. Some studies (e.g., Sinha, 1989) suggest minimal effect of temperature on elasticity, and that, to a good approximation, the elastic stiffness of ice can be assumed to be independent of temperature. In the present study, we adopt this assumption of thermal independence for the elastic moduli. Should temperature dependence be introduced, it must be done with particular care, ideally beginning with a temperature-dependent free energy formulation, to ensure thermodynamic consistency.

Substituting Eq. (6) into (4) gives the familiar isotropic stress-strain relationship:

$$\boldsymbol{\sigma} = (1 - D)\boldsymbol{\sigma}^* = (1 - D) K \varepsilon^e_{vol} \mathbf{I} + 2(1 - D) G \boldsymbol{e}^e \tag{7}$$

## 3.1 Stress-dependent viscosity

The viscous behavior of ice is known to be non-Newtonian (Nye, 1953; Glen, 1955). Herein, we adopt the well-known Glen–Nye flow law's for glacial ice that postulates a power law expression for the creep of polycrystalline ice based on compression tests on ice specimens. Following previous studies (Nye, 1957; Kenny, 1992; Xiao and Jordaan, 1996; Alaei et al., 2021), we assume that the viscous strain is only deviatoric and co-linear with the deviatoric stress. Moreover, following the assumption of Dansereau et al. (2016), the viscosity is assumed to degrade with a factor of $(1 - D)^\alpha$ with $\alpha > 1$ being a model constant, to ensure that the relaxation time decreases with damage. Therefore, the original Glen–Nye flow law can be extended to multiaxial conditions to read (Nye, 1953):

$$\dot{\boldsymbol{\varepsilon}}^v = \dot{\boldsymbol{e}}^v = (1 - D)^\alpha A \exp\left( -\frac{Q}{RT} \right) q^{n-1} \boldsymbol{s} \tag{8}$$





where $A$ is a material constant, $Q$ is the activation energy for creep, $R$ is the universal gas constant, $T$ is the absolute temperature, $n$ is a model exponent, $\boldsymbol{s}$ is the deviatoric stress tensor and $q$ is the deviatoric stress, defined in Sect. 2. The exponent $\alpha > 1$ distinguishes between the rate of degradation in elastic moduli and viscosity, with the former decreasing by $(1-D)$ while latter decreases with $(1-D)^\alpha$. The temperature dependency in Eq. (8) is given by an Arrhenius-type expression, which can be translated as temperature and stress (or rate) dependent viscosity, $\eta$ expressed as:

$$\eta = (1-D)^\alpha \frac{1}{A} \exp\left(\frac{Q}{RT}\right) q^{1-n} = (1-D)^\alpha \frac{1}{A^{\frac{1}{n}}} \exp\left(\frac{Q}{nRT}\right) (\dot{\varepsilon}_q^v)^{\frac{1-n}{n}} \tag{9}$$

with $\dot{\varepsilon}_q^v$ being the rate of the deviatoric strain defined in Sect. 2. The second part of the equality, expressing viscosity in terms of the strain rate, can be derived recalling the co-linearity of the two tensors $\dot{\boldsymbol{e}}^v$ and $\boldsymbol{s}$ implied by Eq. (8). For future purposes, we denote by $g(T)$ and $h(\dot{\varepsilon}_q^v)$ the functions expressing the temperature- and rate-dependency, or *thermokinetic*, of viscosity:

$$\eta = (1-D)^\alpha \, g(T)^{\frac{1}{n}} \, h(\dot{\varepsilon}_q^v)$$

$$g(T) = \frac{1}{A} \exp\left(\frac{Q}{RT}\right), \qquad h(\dot{\varepsilon}_q^v) = (\dot{\varepsilon}_q^v)^{\frac{1-n}{n}} \tag{10}$$

## 3.2 Damage initiation criterion

Experimental results show that the failure strength of ice exhibits thermobarokinetic dependency where it depends not only on confining pressure but also on temperature and strain rate (Jones, 1982; Kalifa et al., 1992; Rist and Murrell, 1994; Gagnon and Gammon, 1995; Mizuno, 1998; Meglis et al., 1999). The effect is presented in Figs. 2 (a) and (b) where the maximum deviatoric stress is plotted versus the corresponding mean stress during triaxial compression tests performed by Jones (1982) and Gagnon and Gammon (1995). This peak deviatoric stress is often interpreted as the threshold beyond which damage starts. A number of key observations are derived from these experiments:

- The failure (damage initiation) envelope is non-linear.

- The threshold is thermobarokinetic.

- Both the shear strength and its pressure-dependency seem to vanish at lower strain rates.

- While not presented here explicitly, a brittle failure is reported for tests at lower mean pressures and higher strain rates, which switches to ductile when pressure is increased and/or strain rate is decreased.

These findings highlight the intertwined effects of strain rate, confining pressure, and temperature in determining both the strength of ice and its transition from brittle to ductile failure. Moreover, at high confining pressures, typically exceeding 40–50 MPa, the deviatoric strength of ice decreases with further increases in the mean pressure. This phenomenon is attributed to pressure melting and dynamic recrystallization, which occur as a result of stress concentration at grain boundaries (Jones, 1982; Jones and Chew, 1983; Rist and Murrell, 1994; Jordaan et al., 1999; Barrette and Jordaan, 2001).

Various yield and failure criteria have been proposed to improve the accuracy of ice failure strength prediction (e.g., Nadreau and Michel, 1986; Fish, 1992; Nadreau et al., 1991; Derradji-Aouat, 2000, 2003). In this study, we adopt an elliptical failure



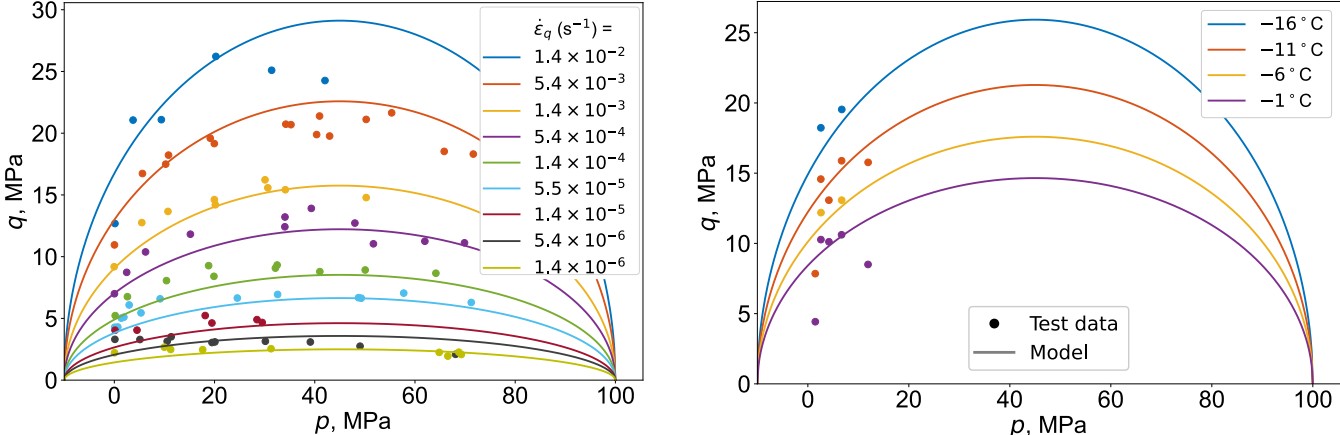

**Figure 2.** Variation of the peak deviatoric stress with confining pressure from triaxial compression tests on ice; (a) data from Jones (1982) at $T = -12\,^{\circ}\text{C}$ at various strain rates, and (b) data from Gagnon and Gammon (1995) showing the effect of temperatures.

criterion similar to the yield surface in Cam-Clay models, which has also been used previously by studies such as the one
by Derradji-Aouat (2000), and can be expressed in the $q - p$ stress space as:

$$\left(\frac{q}{q_{\max}}\right)^2 + \left(\frac{p - p_0}{p_c}\right)^2 = 1 \tag{11}$$

where $p_0$ and $p_c$ are model constants representing the mean stress at ellipse center and major radius of ellipse, respectively, and $q_{\max}$ is the minor radius of the ellipse. Any effect of the third stress invariant (Lode angle) is ignored in this study.

The failure criterion developed by Derradji-Aouat (2000) accounts also for the influence of temperature and strain rate,
and confining pressure on ice strength through making the parameter $q_{\max}$ in Eq. (11) depend on temperature and strain rate. We adopt the same strategy but avoid compounding model parameters by assuming the temperature-, and rate-dependency (thermokinetics) of $q_{\max}$ be similar to that of the viscosity described by Eq. (10), as follows:

$$q_{\max}(T, \dot{\varepsilon}) = q_{\max}^0 \left(g(T)\right)^{1/n} \left(h(\dot{\varepsilon}_q)\right)^{\frac{1}{1-n}} = q_{\max}^0 \frac{1}{A^{\frac{1}{n}}} \exp\left(\frac{Q}{nRT}\right) \dot{\varepsilon}_q^{\frac{1}{n}} \tag{12}$$

where $q_{\max}^0$ is a dimensionless model parameter. Note the slight difference that Eq. (12) depends on the total strain rate,
while Eq. (10) is in terms of the viscous strain rate. The reason is that using either viscous or elastic strain rates will lead to the unrealistic vanishing of $q_{\max}$ either at the beginning of the deviatoric loading ($q = 0 \rightarrow \dot{\varepsilon}_q^v = 0$), or at the steady state ($\dot{q} = 0 \rightarrow \dot{\varepsilon}_q^e = 0$). Nevertheless, following the principle of parsimony in model building, Eq. (12) renders the damage initiation, and the peak deviatoric stress thermokinetic without introducing any new model parameters. Our assumption that the viscous and damage mechanism share in their thermokinetic nature is consistent with previous observations that point to a correlation
between the two mechanism, and more specifically a correlation between the minimum strain rate in a creep test and the failure strength in a constant strain rate test (Mellor and Cole, 1982; Sinha et al., 1995; Barrette and Jordaan, 2003). The form of



Eq. (12) implies that the shear strength vanishes as the strain rate approaches zero, which is consistent with the observations made from the experimental data in Fig. 2.

Restricting the thermokinetics to $q_{\text{max}}$ in Eq. (11) also implies that the minimum and maximum mean pressures (vertices along the major axis) remain independent of strain rate and temperature, which is more or less consistent with the general trends observed in the data in Fig. 2. The maximum mean stress is assumed to depict the pressure melting phenomenon, which is often considered to be independent of strain rate and temperature. The same is assumed to be applicable to the minimum mean stress describing the tensile hydrostatic strength of ice. Figures 2 (a) and (b) showcase the capability of the proposed damage initiation model to capture the experimental trends, with the calibrated parameters presented later on in Table 1. Interestingly, 195  the model performs reasonably well in capturing the thermokinetic damage initiation without resorting to additional calibration parameters. Further exploration of the model in later sections also demonstrate how the transition between brittle and ductile failure is captured by a combination of the Maxwell configuration and the rate-dependency of damage initiation.

    It is worth mentioning that introducing such a rate-dependency directly into the yield surfaces that are subject to consistency condition ($\dot{f} = 0$) requires non-trivial constitutive and numerical treatment, as discussed by Voyiadjis and Abed (2006), among 200  others. The discussion, however, is beyond the scope of the current study.

### 3.3   Energy-based damage evolution law

The model by Dansereau et al. (2016) adopts a stationary damage function independent of $D$, which requires explicit introduction of damage evolution law for the damage parameter. A more consistent alternative is to introduce the dependency on $D$ into the damage initiation function to arrive at a formulation similar to the yield function in elastoplasticity framework, and 205  derive the damage evolution law from the consistency requirement (Simo and Ju, 1987).

    Drawing an analogy with elastoplasticity theory, we define a yield function $f$ in terms of $D$ and its conjugate $Y$ which is based on the simple general form proposed by Marigo (1982):

$$f := Y - \kappa(\kappa_0, D) \le 0$$
$$\kappa(\kappa_0, D) = \kappa_0(1 + mD) \tag{13}$$
$$\kappa_0(\boldsymbol{\sigma}^*, T, \dot{\varepsilon}) = \frac{p^{*2}}{2K} + \frac{q_{\text{max}}^2(T, \dot{\varepsilon}_q)}{6G}\left(1 - \left(\frac{p^* - p_0}{p_c}\right)^2\right)$$

where $K = \frac{E}{3(1-2\nu)}$ and $G = \frac{E}{2(1+\nu)}$ are the undamaged elastic bulk and shear moduli, $\boldsymbol{\sigma}^*$ is the undamaged stress tensor 210  with $p^*$ being its mean stress, $\kappa_0$ represents the threshold energy, independent of damage variable $D$, required to initiate damage growth, and $m \ge 0$ is a model constant. The expression for $\kappa_0$ in Eq. (13) is derived by noting that at $D = 0$, the expression of $f$ should revert to the damage initiation function given in Eq. (11), and recalling that for isotropic elasticity, $Y = (p^*)^2/(2K) + (q^*)^2/(6G)$. The condition $f \le 0$ requires that during yielding, the state of the material remains on the $f$, which is described by the consistency equation $df = 0$:

$$df = \frac{\partial f}{\partial Y}\frac{\partial Y}{\partial \boldsymbol{\sigma}^*} : d\boldsymbol{\sigma}^* - \frac{\partial \kappa}{\partial \boldsymbol{\sigma}^*} : d\boldsymbol{\sigma}^* - \frac{\partial \kappa}{\partial D}dD - \frac{\partial \kappa}{\partial \dot{\varepsilon}_q}d\dot{\varepsilon}_q - \frac{\partial \kappa}{\partial T}dT \tag{14}$$



where $d\dot{\varepsilon}_q$ denotes a change in the deviatoric strain rate. Recalling $d\boldsymbol{\sigma}^* = \mathbb{C}^e : d\boldsymbol{\varepsilon}^e$, Eq. (14) can be rearranged to calculate the damage evolution $dD$ that is required for the material state to remain on the yield surface:

$$dD = (\frac{\partial \kappa}{\partial D})^{-1} \left( \frac{\partial f}{\partial Y} \frac{\partial Y}{\partial \boldsymbol{\sigma}^*} - \frac{\partial \kappa}{\partial \boldsymbol{\sigma}^*} \right) : \mathbb{C}^e : d\boldsymbol{\varepsilon}^e - (\frac{\partial \kappa}{\partial D})^{-1} \left( \frac{\partial \kappa}{\partial \dot{\varepsilon}_q} d\dot{\varepsilon}_q + \frac{\partial \kappa}{\partial T} dT \right) \geq 0 \tag{15}$$

which can be rewritten for clarity as:

$$dD = \mathbf{M} : d\boldsymbol{\varepsilon}^e - N$$

$$\mathbf{M} = (\frac{\partial \kappa}{\partial D})^{-1} \left( \frac{\partial f}{\partial Y} \frac{\partial Y}{\partial \boldsymbol{\sigma}^*} - \frac{\partial \kappa}{\partial \boldsymbol{\sigma}^*} \right) : \mathbb{C}^e, \quad N = (\frac{\partial \kappa}{\partial D})^{-1} \left( \frac{\partial \kappa}{\partial \dot{\varepsilon}_q} d\dot{\varepsilon}_q + \frac{\partial \kappa}{\partial T} dT \right) \tag{16}$$

The first term in Eq. (16) represents the damage evolution due to change in elastic strain while the second term accounts for the change in damage level as a result of change in strain rate and temperature. Kuhn–Tucker condition requires that damage evolution (yielding) occurs when $\frac{\partial f}{\partial Y} dY = dY > 0$, otherwise, the damage remains constant, unless healing is activated, as discussed in Sect. 5.

Equation (16) can now be substituted into the incremental form of Eq. (4), which, together with the total strain partitioning for Maxwell configuration, $d\boldsymbol{\varepsilon}^e = d\boldsymbol{\varepsilon} - \frac{1}{\eta}\boldsymbol{s}dt$, leads to the following general incremental stress-strain relation:

$$d\boldsymbol{\sigma} = \mathbb{C}^D : d\boldsymbol{\varepsilon} - (\frac{1}{\eta}\mathbb{C}^D : \boldsymbol{s}dt + \mathbb{C}^e : \boldsymbol{\varepsilon}^e N_D)$$

$$\mathbb{C}^D = \mathbb{C}^e : \left[ \left( (1-D)\mathbf{I} \otimes \mathbf{I} - \boldsymbol{\varepsilon}^e \otimes \mathbf{M}_D \right) \right] \tag{17}$$

with

$$\mathbf{M}_D = \mathbf{M} \quad \text{and} \quad N_D = N, \quad \text{for } dY > 0$$

$$\mathbf{M}_D = \mathbf{0} \quad \text{and} \quad N_D = 0, \quad \text{otherwise.} \tag{18}$$

The second expression in Eq. (18) also describes load reversal (unloading) under constant $D$.

## 4  Model validation and parametric exploration

The incremental form of the constitutive model is described in its entirety by Eqs. (17), (18), (16), (13), (12), (8), and (6). The system of equations are solved using explicit numerical integration method.

The model parameters are calibrated using multiple datasets on triaxial compression of ice; the parameters introduced for the damage initiation function, i.e. $Q$, $n$, $A$, $p_0$, $p_c$ and $q_{\max}^0$ in Eqs. (11) and (12), were calibrated using the datasets from Jones (1982) and Gagnon and Gammon (1995) with the predictions already presented in Fig. 2. Chemically meaningful variables, such as $Q$ are kept within previously reported ranges. The value of the exponent $n$ is also within the range of experimental observations compiled by Weertman (1983). The rest of the model parameters, $E$, $\nu$, and $m$, describing the transient response with strain, are calibrated based on the triaxial dataset by Rist and Murrell (1994) who performed triaxial compression tests at a constant confining pressure of $10\,\mathrm{MPa}$, with strain rates ranging from $1 \times 10^{-5}\,\mathrm{s}^{-1}$ to $1 \times 10^{-2}\,\mathrm{s}^{-1}$ at a temperature of



$-20\,°\mathrm{C}$. The elastic parameters, $E$ and $\nu$, are calibrated directly based on the instantaneous stress-strain slopes, while the combination of parameters $A$ and $Q$ were fine-tuned based on the variation of the steady state stress with strain rate. The remaining parameter $m$ controlling the damage evolution is calibrated based on the post-peak softening rate. The calibrated material parameters are compiled in Table 1. It should be noted that, in the absence of careful theory-led experiments, the measurement of

245 elastic moduli becomes subjective to the assumed rheological configuration; for instance, the Maxwell configuration assumed herein excludes the time-dependent reversible deformation sometimes introduced as primary creep, which can lead to a lower calibrated Young's modulus in comparison to the reported values of $E = 9.7\text{-}11.2\,\mathrm{GPa}$ (Sinha, 1989; Petrovic, 2003).

**Table 1.** Calibrated model parameters

| $E$ | $\nu$ | $A$ | $Q$ | $n$ | $p_0$ | $p_c$ | $q_{\max}^0$ | $m$ | $\alpha$ |
|---|---|---|---|---|---|---|---|---|---|
| (GPa) | – | $(\mathrm{s}^{-1}\,\mathrm{MPa}^{-n})$ | $(\mathrm{kJ\,mol}^{-1})$ | – | (MPa) | (MPa) | – | – | – |
| 4.6 | 0.33 | $9.48 \times 10^8$ | 83 | 3.75 | 55 | 45 | 0.84 | 1.2 | 4.8 |

Figure 3 compares the model predictions with the experimental results with different strain rates, where an acceptable accuracy is observed in capturing both the maximum and residual deviatoric stresses as they vary with the imposed strain rate.

The transition from ductile to brittle failure as a result of higher strain rates is also appropriately represented in the model. The evolution of the damage parameter is also given as an inset in Fig. 3 which indicates the effect of strain rate on damage initiation strains and the eventual damage value at steady state. Referring back to Eq. (9), the viscosity scales with $q^{1-n}$, which together with the calibrated value of $n > 1$, indicates that the viscosity becomes infinity at the start of the loading when $q = 0$. This implies that the instantaneous response is elastic. The viscous mechanism becomes gradually more significant as

$q$ increases, and particularly after the damage mechanism starts. The degradation of viscosity with damage causes the viscous mechanism absorb more share of the imposed deformation rate, until a balance is reached at a particular damage level whereby at a constant strain rate, all the imposed deformation is accommodated by the viscous mechanism which remains at equilibrium with the stationary stress induced within the elastic mechanism.

For further validation, the calibrated model is used, without any additional adjustments, to predict a separate set of triaxial

experiments conducted by Murrell et al. (1991).These tests were performed under confining pressures ranging from $10\,\mathrm{MPa}$ to $30\,\mathrm{MPa}$, at a fixed strain rate of $1 \times 10^{-2}\,\mathrm{s}^{-1}$, and at the same temperature of $-20\,°\mathrm{C}$ as in Rist and Murrell (1994). Figure 4 shows the stress-strain results for different confining stresses. While the lack of direct calibration causes the model to be less accurate compared to the results in Fig. 3, nevertheless, the model shows a plausible performance in capturing the peak stress and the pressure dependency of the failure nature, as it transitions from brittle to ductile for larger confining stresses.

Compared to the experimental data, the model exhibits a higher sensitivity of the peak stress on the confining pressure, which, we conjecture, arises from the elliptical failure envelope used for peak stress prediction. The additional sensitivity of the residual stress is probably a secondary effect caused by the extra sensitivity of the peak stress. The evolution of the damage parameter $D$ is also presented in Fig. 4 as an inset. Unlike that in Fig. 3, the fixed strain rate causes the damage mechanism



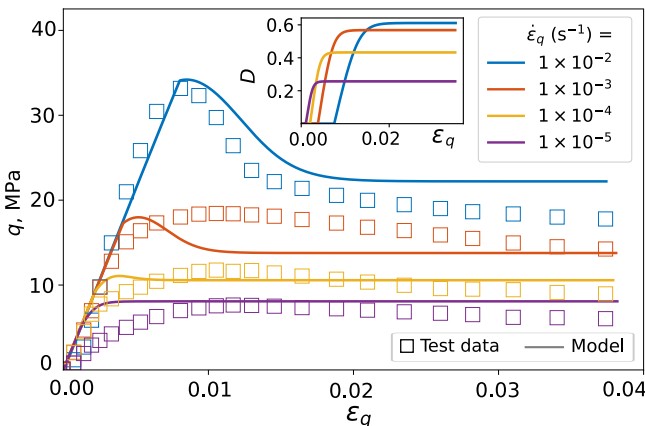

**Figure 3.** Comparison between the calibrated model results and triaxial compression data from Rist and Murrell (1994) at $-20\,^\circ$C and $10\,$MPa confining pressure for various strain rates. The inset shows the associated change in the damage parameter.

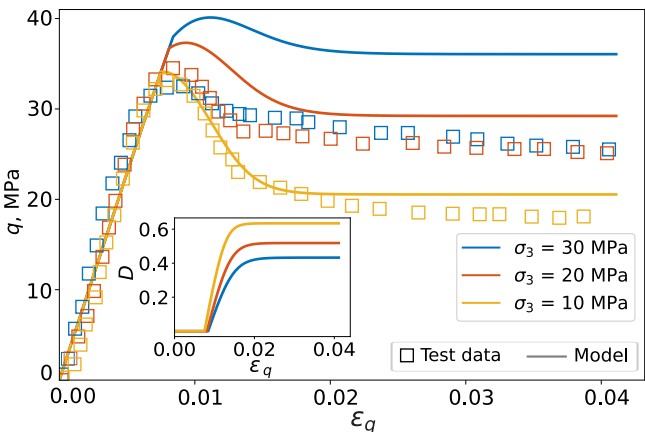

**Figure 4.** Comparison between calibrated model results and triaxial compression data from Murrell et al. (1991) at $-20\,^\circ$C with a strain rate of $\dot{\varepsilon}_q = 1 \times 10^{-2}\,\mathrm{s}^{-1}$ under different levels of confining pressure. The inset shows the associated change in the damage parameter.

to start at relatively the same strain level. The model successfully captures the well-known phenomenon of damage growth
suppression, leading to lower level of damage, and hence more ductility, at higher pressures.

To better visualize the rate-dependent characteristics of the model, the same calibrated model has been used to simulate a relaxation test under uniaxial condition where the deviatoric stress is first increased to an initial value of $(\sigma_1)_0$ at $T = -3\,^\circ$C. The stress is then allowed to relax under fixed strains. The temperature and the stress conditions are chosen to resemble the conditions reported in the study by Voitkovsky (1967). The relaxation of the stress for the two initial deviatoric stresses is
shown in Fig. 5. The general trends of stress relaxation resembles those reported by Voitkovsky (1967). The half-relaxation times (i.e., time required for stress to drop to $(\sigma_1)_0/2$) are found to be $2.2\,$h and $0.89\,$h for the cases with $(\sigma_1)_0 = 0.5\,$MPa



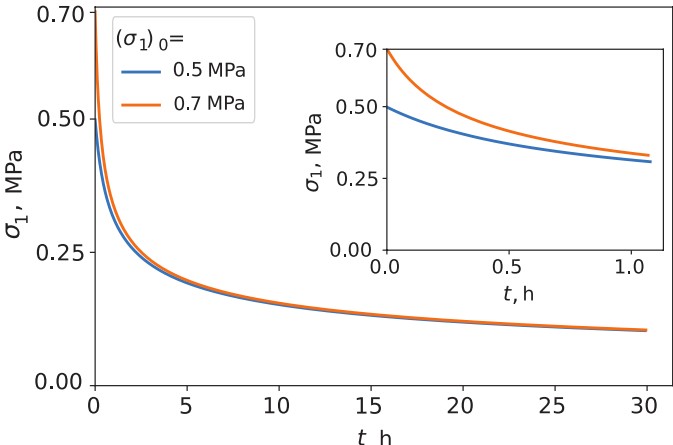

**Figure 5.** Relaxation of deviatoric stress under fixed strains, uniaxial condition at $T = -3\,^{\circ}\mathrm{C}$, similar to the experiments reported by Voitkovsky (1967). The inset figure shows a zoomed view of the first hour.

and $0.7\,\mathrm{MPa}$, respectively, which are in general agreement with the ranges of $0.58$-$1.7\,\mathrm{h}$ and $1.6$-$3.2\,\mathrm{h}$ reported by Voitkovsky (1967) for the same stress levels.

The rate-dependency of the model makes it difficult to properly simulate a stress-controlled loading, similar to creep tests, where the change in strain rate ($d\dot{\varepsilon}_q$ required for the consistency equation) continuously evolve and is unknown. This issue, however, does not prevent implementation of the model in computational boundary value problem solvers that are often formulated in terms of imposed strains and strain rates. A constant stress can be achieved through a servo-control mechanism, which has not been considered in this study. However, if a constant strain rate is achieved under isothermal constant stress conditions, then the damage level will be stable and the $d\boldsymbol{\sigma} = 0$ will imply that the total strain rate is equal to the viscous strain rate, which is described by Eq. (8), which indicates a scaling of $d\dot{\varepsilon}_q \propto q^n$. This echoes the previous studies on the tertiary creep of ice whereby a similar scaling was reported (Paterson, 1977; Weertman, 1983; Treverrow et al., 2012), with an exponent of $n = 3.8 \pm 0.3$ reported by Treverrow et al. (2012), which includes the value of $3.75$ obtained through our calibration in Table 1. Nevertheless, the Maxwell configuration used in the current model is known to produce a two-stage creep response under constant stress conditions, as opposed to the three-stage creep behavior commonly recognized in the literature (Paterson, 1977).

## 4.1 Parametric expolorations

As a parametric study, the effect of varying temperatures on the stress-strain predictions is visualized in Fig. 6 for triaxial compression conditions with a confining pressure of $10\,\mathrm{MPa}$ and a constant strain rate of $\varepsilon_q = 1 \times 10^{-3}\,\mathrm{s}^{-1}$. With the inclusion of temperature dependency into both the damage yield function and the viscosity formulation, the model captures the general observed trends of increase in strength and the decrease in ductility for lower temperatures. It is worth noting that the accuracy of the model is expected to deteriorate for temperate ice and ice near its melting point. In this regime, the microstructure of ice



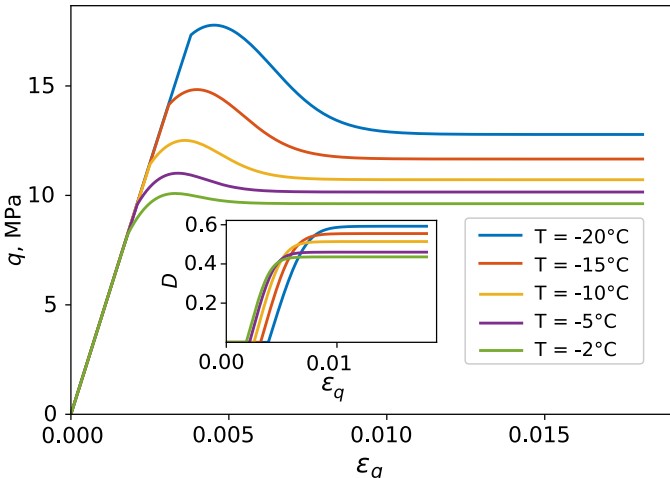

**Figure 6.** Effect of temperature on predicted stress–strain response of triaxial compression under $10\,\mathrm{MPa}$ confining pressure and a constant strain rate of $\varepsilon_q = 1 \times 10^{-3}\,\mathrm{s}^{-1}$. The inset shows the associated change in the damage parameter.

undergoes significant alterations, and the relationship between strain rate and temperature deviates from a simple Arrhenius-type relation used for both viscous and damage mechanisms (Mellor and Testa, 1969; Morgan, 1991; Cole, 2020).

The effect of varying strain rate on the deviatoric stress is visualized in Fig. 7 for a triaxial compression test where the axial strain rate is increased in two steps from $\dot{\varepsilon}_1 = 1.5 \times 10^{-5}\,\mathrm{s}^{-1}$ to $1.5 \times 10^{-3}\,\mathrm{s}^{-1}$. The higher strain rate induces higher deviatoric stresses through the combined effect of a higher viscous stress and the expansion of the damage yield function. The response also becomes more brittle at higher strain rates, as is expected. The evolution of damage parameter is also presented as an inset in Fig. 7 where the increase of damage with strain is a further indicator of the increased brittleness. Note that the rate-dependency of the damage mechanism and the inclusion of $d\dot{\varepsilon}_q$ in the consistency relation (Eq. (14)) prevents the possibility

of jumps in strain rates, and as a result, short periods of smooth transitions are included to ramp up the strain rate.

To better demonstrate the capability of the proposed model in capturing complex loading paths, we consider a condition where the temperature is changing while the material is under constant strain rate. The variation in the deviatoric stress and temperature with time is shown in Fig. 8 where the temperature is increased from $T = -20\,^{\circ}\mathrm{C}$ to $-5\,^{\circ}\mathrm{C}$ under a constant strain rate of $\dot{\varepsilon}_q = 1.5 \times 10^{-3}\,\mathrm{s}^{-1}$. As expected, the deviatoric stress gradually decreases as the temperature raises, indicating

a degradation in ice strength. The inset of Fig. 8 shows the variation of the damage parameter, which, interestingly, does not further increase when the temperature is increased and the deviatoric stress drops. This is due to the sample becoming more ductile at higher temperatures, which precludes further damage. Instead, the stress states glides along the yield surface that is shrinking as the temperature is increased.





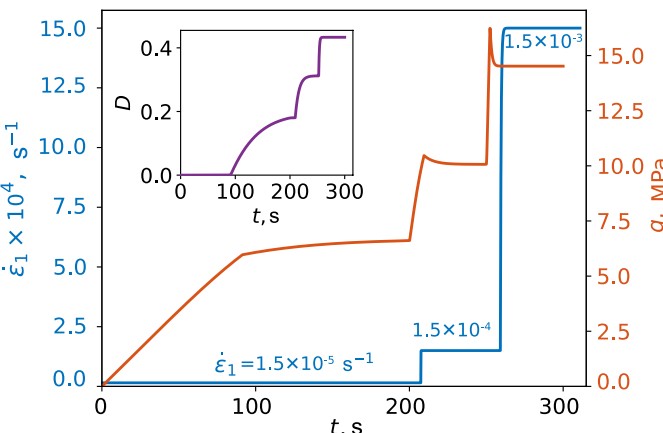

**Figure 7.** Deviatoric stress response of a triaxial compression under $\sigma_3 = 10\,\text{MPa}$ and a constant temperature of $T = -20\,^\circ\text{C}$. Axial strain rate is increased in two steps from $\dot{\varepsilon}_1 = 1.5 \times 10^{-5}\,\text{s}^{-1}$ to $1.5 \times 10^{-3}\,\text{s}^{-1}$. The inset shows the associated change in the damage parameter.

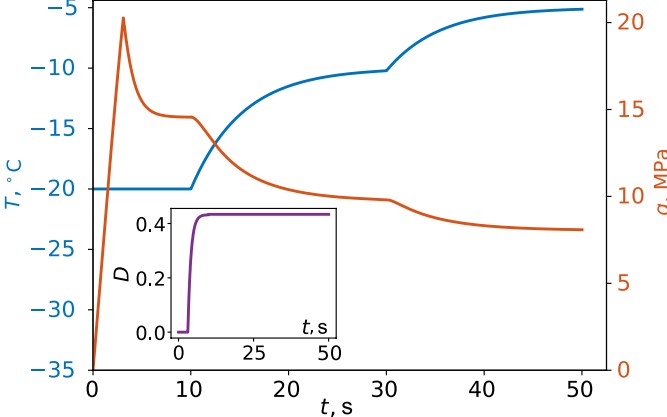

**Figure 8.** Deviatoric response of a triaxial compression under $\sigma_3 = 10\,\text{MPa}$ and a constant strain rate of $\dot{\varepsilon}_q = 1.5 \times 10^{-3}\,\text{s}^{-1}$, while the temperature is increased gradually from $T = -20\,^\circ\text{C}$ to $-5\,^\circ\text{C}$. The inset shows the associated change in the damage parameter.





## 5 Healing mechanism

Ice, like many crystalline materials, particularly those with a granular structure, exhibits the capacity to heal after fracturing, regaining strength and stiffness over time (Miao et al., 1995). The healing of ice is an active area of research and is generally regarded as a thermally activated process, driven primarily by surface energy minimization through mass transport (Blackford, 2007). Among the proposed mechanisms, sintering via surface diffusion is considered the dominant mode of crack healing (Murdza et al., 2022). Additionally, the presence of a liquid-like layer on crack surfaces is thought to further enhance the 320 healing process (Demmenie et al., 2022, 2025).

In their model for sea-ice, Dansereau et al. (2016) adopted a time-driven healing with an imposed constant rate. However, the healing through sintering is reported to happen during short periods as a result of impact load during collision of ice particle (Szabo and Schneebeli, 2007; Bahaloo et al., 2022). Moreover, the healing rate is reported to be affected by temperature and pressure (Schulson et al., 2016). Motivated by these observation, we put forward a new model for healing of ice whereby the 325 the energy absorbed into the viscous mechanism is assumed to be deriving the healing. The energy dissipated as heat through the viscous processes can be regarded as a localized heat source enhancing the molecular mobility at the crack interface, and accelerating surface diffusion of water molecules and promoting the formation of a thicker liquid-like layer, both of which contribute to enhanced healing (Murdza et al., 2022). The proposed mechanism is similar to the viscous sintering observed in amorphous solids such as glass and magma (Scherer and Garino, 1985; Vasseur et al., 2013). We propose that the rate of 330 healing $\dot{D}_h$ is proportional to the viscous energy release, as expressed by:

$$\dot{D}_h = -\frac{\boldsymbol{s} : \dot{\boldsymbol{e}}^v}{E_h} = -\frac{\boldsymbol{s} : \boldsymbol{s}}{\eta\,E_h} = -\frac{2q^2}{3\eta\,E_h} \tag{19}$$

where $\dot{e}_{ij}^v$ is the viscous strain rate and $E_h$ is an additional model constant representing the internal characteristic energy required for healing. Being expressed in terms of the viscous energy, the healing rate naturally inherits its pressure and temperature dependency.

Including healing into the damage mechanics, however, is not straightforward, since the damage yield function, together with the consistency requirement (Eq. (14)), fully describes the evolution of the damage level, and hence, the healing cannot be introduced into the yield function. Within the framework of damage mechanics, the issue is handled by introducing two distinct state variables for damage and healing, whose combined effect is included in the free energy potential. This strategy seems to be less relevant for ice since damage and healing seem to have similar and opposite effects that should be described by 340 a single state variable, as proposed by Dansereau et al. (2016). Herein, we sidestep the issue by activating the healing process, described by Eq. (19), only when the state of the damaged material is below the yield limit, i.e. $f < 0$, and assume that the damage evolution given by Eq. (14) describes the combined effect damage and healing. Therefore, when $f < 0$ the elasticity formulation will be replaced with

$$d\boldsymbol{\sigma} = \mathbb{C}^e : \left[(1-D)d\boldsymbol{\varepsilon}^e - \boldsymbol{\varepsilon}^e dD_h\right] = \mathbb{C}^e : \left[(1-D)d\boldsymbol{\varepsilon}^e + \boldsymbol{\varepsilon}^e \left(\frac{2q^2}{3\eta\,E_h}\right)dt\right] \tag{20}$$

with the second term describing the increase in stress due to a decrease in the damage level as a result of healing described by Eq. (19). It can be easily shown that the Maxwell system will be thermodynamically admissible and energy creation is





prevented as long as the the characteristic healing energy is greater than the undamaged elastic energy, i.e. $Y < E_h$. Naturally, the healing process is only activated when $D > 0$.

The effect of the proposed healing mechanism is demonstrated in Fig. 9 which presents the deviatoric stress-strain response

and the damage evolution, with and without healing mechanism, in a triaxial test under conditions similar to those for Fig. 8, with $T = -20\,^\circ\mathrm{C}$ and an arbitrarily chosen value of $E_h = 5\,\mathrm{MPa}$ for the healing process. The loading involves an initial constant strain rate of $\dot{\varepsilon}_1 = 1.5 \times 10^{-3}\,\mathrm{s}^{-1}$ followed by gradually decreasing and reversing the strain rate until $q = 0$, and reloading with gradually increasing strain rate to its initial value. During the unloading stage and stress relaxation, the state of the material falls below the yield function ($f < 0$) which activates the healing mechanism. The energy stored in the elastic

component is gradually transferred to the viscous component, a portion of which will reverse the damage through the viscous healing process described by Eq. (19). Figure 9 shows that, when healing is activated, the damage level (dashed line) starts to decrease during the unloading stage until almost half of the damage incurred during the first loading stage is recovered. Upon reloading, the damage level starts to rise again approaching the eventual value achieved by the simulation without healing (solid line). When healing is present, the associated stress-stress response during the unloading-reloading cycle exhibits a higher

stiffness, which can be attributed to the recovery of the elastic stiffness. This higher stiffness induced by healing qualitatively resembles the response reported in loading/unloading tests on ice (see Kenny (1992)).

Experimental data targeting the healing of ice at the macroscopic scale within the framework of damage mechanics is indeed scarce. Therefore, while the formulation of healing given in Eq. (19) appears to capture qualitatively the basics of the healing process, its quantitative performance and its natural dependence on parameters such as pressure and temperature remain to be

verified. Further exploration of these effects is left to future studies. Nevertheless, the present formulation offers an appropriate first step toward incorporating healing into constitutive models of ice. Including such mechanisms is essential for accurately capturing long-term mechanical behavior of ice where damage reversal can significantly influence strength, stiffness, and structural integrity.

## 6 Conclusion

This study presented a comprehensive three-dimensional constitutive model for ice, developed to capture the intricate and interdependent effects of temperature, pressure, and strain rate, referred to as the thermobarokinetics of ice. Building upon the Maxwell elasto-brittle framework of Dansereau et al. (2016), the proposed model introduces several key enhancements that address critical limitations in existing approaches. As a central contribution is the adoption of a unified Arrhenius-type expression for temperature and pressure dependence of both the viscosity and the damage initiation threshold, which allows the

model to reproduce the interdependence of thermal, mechanical, and kinetic effects using a minimal and physically meaningful set of parameters.

The model was calibrated and tested against independent datasets from triaxial compression experiments on polycrystalline ice. Validation results confirmed the model's ability to replicate a broad spectrum of observed behaviors. The model plausibly reproduces the stress–strain response of ice, including the transition from ductile to brittle failure as a function of strain rate





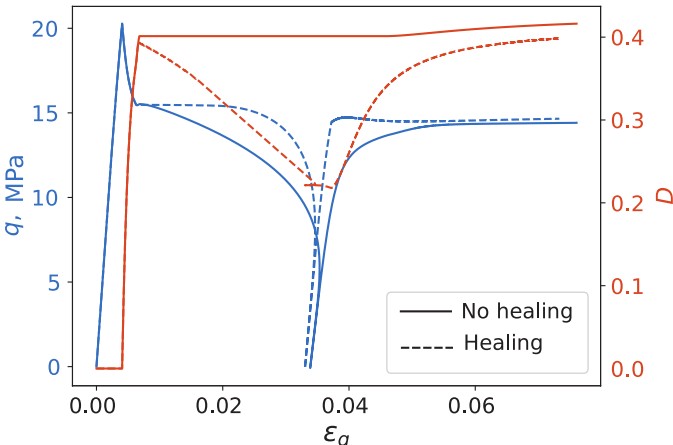

**Figure 9.** Deviatoric stress-strain response and damage evolution during a unloading cycle with and without the healing mechanism. The simulation parameters are similar to those in Fig. 8 with $T = -20\,^\circ$C and $E_h = 5$ MPa for healing model.

and confining pressure, and captures both peak and residual stress magnitudes with relatively good accuracy in line with experimental observations. The model also reflects the suppressive role of high confining pressure on damage development, as well as the sensitivity of ice strength and ductility to temperature. Moreover, its predictions for stress relaxation align closely with available time-dependent experimental data, further reinforcing its credibility.

A physically motivated formulation for healing was also proposed, inspired by viscous sintering processes observed in amorphous materials such as glass. This mechanism ties the rate of healing to the rate of viscous energy dissipation, thereby embedding pressure and temperature sensitivity into the recovery process. Preliminary simulations indicate a progressive restoration of stiffness during relaxation, offering a promising path forward for modeling long-term behavior and damage recovery in ice.

Despite these achievements, certain limitations remain. The model's predictive accuracy diminishes for temperate ice near its melting point, where the assumed Arrhenius-type temperature dependence may no longer be valid. Additionally, incorporating strain rate and temperature into the yield function without accounting for their thermodynamic conjugates limits the model's applicability to scenarios where these parameters are externally controlled.

Finally, while the proposed healing formulation offers a coherent first step, it requires rigorous experimental validation, particularly under varying thermal and mechanical conditions, to establish its quantitative reliability. Addressing these challenges will further enhance the model's applicability to a wide range of cryospheric, glaciological, and geotechnical engineering problems, especially those involving long-term loading, cyclic deformation, or evolving thermal regimes.

*Author contributions.* **F. Sahragard:** Conceptualization, Formal analysis, Methodology, Validation, Visualization, Writing – original draft, Writing – review and editing. **M. Pouragha:** Conceptualization, Formal analysis, Methodology Funding acquisition, Supervision, Validation,



Visualization, Writing – original draft, Writing – review and editing. **M. Rayhani:** Funding acquisition, Resources, Supervision, Writing – review and editing

*Competing interests.* The authors declare that they have no conflict of interest.

*Acknowledgements.* The authors would like to thank Dr. Rachel Glade (University of Rochester) and Dr. Véronique Dansereau (ISTerre-IGE, France) for the engaging and insightful discussions, which significantly helped the development of this work. This study was supported by the Natural Sciences and Engineering Research Council of Canada [RGPIN-2020-06480 and ALLRP-2023-590791 held by M. Pouragha and Discovery Grant held by M. Rayhani]



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
