# Peer review of "Thermobarokinetics of ice: constitutive formulation for the coupled effect of temperature, stress, and strain rate in ice"

_EGUsphere, 2025_

## Author Response (AR1)

**Thermobarokinetics of ice: constitutive formulation for the coupled effect of temperature, stress, and strain rate in ice**
**Response to the Reviewers' Comments**

We sincerely thank the reviewers for their thorough evaluation and valuable feedback on our manuscript. We have carefully addressed every point raised by the reviewers and made significant changes and additions to the manuscript, which have improved the clarity of the main ideas and readability of the paper.

**Reviewer 1**

**Rev 1:** I thoroughly enjoyed reading "Thermobarokinetics of ice: constitutive formulation for the coupled effect of temperature, stress, and strain rate in ice." This paper carefully develops a new multi-part model of ice rheology by building on previously proposed models for the elastic, viscous, and damage/healing response of ice in the presence of varying pressure, temperature and strain rate, with the goal of illuminating coupling among these complex components. I appreciated the way the authors carefully and convincingly develop their model, and their comparison with real experimental data was a nice way to demonstrating both the strengths and limitations of the model. I think this paper is an excellent contribution to our understanding of the mechanics of ice deformation with many implications for glaciology and other icy systems. The presentation, writing, and figures are all very high quality, and the literature review is very thorough

and nicely done. While I don't feel qualified to comment on the accuracy of the presented equations, I had no problem following the logic of the paper and am confident that the equations presented are justified (I would encourage the authors to double check for typos in equations before the final revision). I have only a few minor questions/comments about this paper, outlined below. These are largely suggestions, and I leave it to the authors to decide whether they are worth addressing in the present study or in future work.

We sincerely thank the reviewer for the positive feedback and the helpful suggestions; we have carefully re-checked the equations for any potential typos and inconsistencies.

**Rev 1:** I appreciated the approach of calibrating the model to one set of experiments, and then applying that calibrated model to another set. This is a nice way of testing the universality of the model and sensitivity to calibrated parameters. I imagine different instances of ice have inherent variability in parameters such as A, E and nu, and I am curious whether there is enough information in the literature to constrain a reasonable range of these parameters in natural systems. Do they range over orders of magnitude? Are there any physically based ways of estimating them? This is largely beyond the scope of this paper, but a little bit of information in the discussion might be nice.

We thank the reviewer for raising this important point. To the best of our knowledge, currently there is no standard calibration protocol for triaxial ice rheology. Moreover, any calibration program, and the calibrated values, will depend on the assumed constitutive model. For instance, assuming Maxwell vs. Kelvin configuration leads to different calibration protocols and potentially different calibrate parameters. On the other hand, the existing experimental datasets on ice mechanics are often exploratory in nature, and not designed for calibrating specific models. With that being said, we tried to resort to a semi-systematic calibration procedure (as explained in the paper) which allowed us to directly calibrate some of the important parameters, such as those for Glen–Nye's law.

The point regarding the reasonable ranges is very valid. We will add the following paragraph to the Discussion section:

"*The parameters in Table 1 are calibrated for isotropic, freshwater polycrystalline ice. Under this assumption the literature indicates a comparatively narrow band for elastic constants, with Sinha (1989) reporting $E \approx 8.93$–$9.53$ GPa and $\nu \approx 0.308$–$0.311$ over 0 to $-50\,°C$. The Maxwell configuration excludes the reversible rate-dependent deformation (primary creep), which is known to lead to lower Young's modulus values in comparison to commonly reported values. For the Glen–Arrhenius set $\{A, n, Q\}$, reported values show bounded variability with $n \approx 2$–$4$, $Q \approx 60$–$120$ kJ mol$^{-1}$, while A spans orders of magnitude across studies due to microstructure and differences in calibration windows (Glen, 1955; Barnes et al., 1971; Zeitz et al., 2020). Within this context, our calibrated $\{A, n, Q\}$ lie within established intervals and are consistent with ranges reported by Durham et al. (1983) for polycrystalline ice.*"

**Rev 1:** Along these lines of relating the paper to real systems, a 1km thick glacier might experience a pressure at the base on order of tens of MPa. This nicely falls within the regime of parameters used in this study, enhancing applicability to glaciology (something the authors might want to mention). This is especially important for studies of glacial erosion, in which the shear stress and velocity exerted on the bed at the bottom of the glacier determine erosion. Again, this is outside the scope of the study but it might be worth mentioning a few key papers to enhance the paper's broad applicability and importance (e.g.,

https://www.nature.com/articles/s43017-021-00165-9;

https://www.nature.com/articles/s41467-020-14583-8;

https://www.science.org/doi/full/10.1126/science.aab2386).

We thank the review for pointing out this connection. We will add the following to the newly added *Limitations and Range of Applicability* section:

"*The triaxial behavior has been compared only at relatively high mean stress levels, on the order of 10 MPa, due to the limited availability of experimental data. These stress magnitudes are particularly relevant as they are comparable to basal overburden pressures typically encountered beneath glaciers (Herman et al., 2015, 2021; Cook et al., 2020).*"

**Rev 1:** It would be nice to include some discussion about what would be the most useful future studies- especially experimental- that could be done to improve our understanding of ice mechanics in light of the findings in this paper. There could be a paragraph or two about this toward the end of the paper.

The following passage will be added as suggested:

"*Dedicated experiments are needed to determine which rheological framework (Maxwell, Kelvin, Burgers, etc.) most closely represents ice behavior. Moreover, although the viscous-driven healing process proposed here is physically appealing, it remains unverified by experiments. Targeted experiments, particularly load–hold–reload and low-frequency cyclic triaxial tests conducted under controlled confinement and temperature, are essential to confirm the potential correlation between strain rate and stiffness recovery.*"

**Reviewer 2**

**Rev 2:** This study presents a constitutive model for ice that captures the complex and interdependent effects of temperature, pressure, and strain rate. A key contribution of the work is the use of a unified Arrhenius-type formulation to describe the temperature- and pressure-dependence of both viscosity and the damage initiation threshold. The model is calibrated and validated using independent datasets from triaxial compression tests on polycrystalline ice. Overall, the paper is clearly written, and both the theoretical formulation and validation are well explained. However, several points should be addressed to further strengthen the contribution and improve the clarity of the work: We thank the reviewer for the detailed assessment of the manuscript.

**Rev 2:** Provide a more detailed discussion of how temperature influences the mechanical properties and overall model response. As an example the authors considered constant elastic modulus and Poisson's ratio. However, in frozen soil, the mechanical properties of the medium vary as a function of temperature or ice saturation.

It seems from this and other comments that the reviewer is comparing the results of our study with frozen soils, most probably because frozen soil literature often involves ice-soil mixtures. However, our model is developed for pure ice and not mixture of ice and soil. Concepts such as ice saturation are only applicable to the latter not the former. The temperature-dependent behavior is already explicitly represented in the model through two mechanisms: (i) the Arrhenius-type viscosity law and (ii) the temperature-dependent damage initiation threshold see Eqs. 9-12. Regarding the elastic constants, we have clarified explicitly that their potential temperature dependency is ignored in this study, see the discussion following Eq. 6. This is because, as explained in the original manuscript, experimental data show that the variation of elastic parameters with temperature is indeed negligible. Sinha (1989) reported that between 233 K and 273 K, the Young's modulus of ice increases by about 5%, and the Poisson's ratio by only 1%, which is negligible compared with the temperature-dependency of viscosity and damage. Therefore, assuming constant $E$ and $\nu$ introduces minimal error while simplifying the analysis.

**Rev 2:** The incremental form of the constitutive model is solved using an explicit numerical integration scheme. Please specify the computational platform or software (e.g., in-house code, ABAQUS, COMSOL, etc.) used for the implementation. Additionally,

discuss whether any stability issues were encountered given the explicit formulation and how these were mitigated.

The model was implemented in an in-house Python code using an explicit integration scheme. No numerical instabilities were encountered, and sensitivity tests confirmed that further step-size reduction produced no meaningful change in the results. The following sentence will be added to the revised manuscript in response to the this comment:

" *The system of equations are solved using explicit numerical integration method. Sensitivity tests were performed to confirm that the selected timesteps were sufficiently small, and further step-size reduction produced no meaningful change in the results.*"

**Rev 2:** Clarify whether the liquid water and ice contents are assumed constant. If the volumetric ratio between the phases changes, how would this affect the governing equations and model predictions?

Is the medium assumed to be fully saturated? If so, please elaborate on how the model would perform or need to be modified under unsaturated conditions.

Similar to the first comment, it seems the developed model for pure ice in this study is being conceived as for frozen soil, which has led to some confusion. The present constitutive formulation describes the thermomechanical behavior of *single-phase polycrystalline ice*. The material is treated as a continuous solid without any soil skeleton, liquid water, or air phases; hence, concepts such as saturation or evolving phase fractions do not apply.

**Rev 2:** A dedicated section discussing the limitations and applicability range of the proposed model would be beneficial.

Thank for this suggestion. We will add the following section to the revised manuscript:

"*The proposed model reproduces the main features of the triaxial response within the calibrated window, including initial stiffness, peak strength, and post-peak softening. The*

*formulation nevertheless relies on simplifying assumptions that limit its applicability. It assumes isotropy, while elasticity, creep, and damage in polycrystalline ice depend on crystallographic fabric and on the alignment of microcracks, which can produce direction-dependent stiffness and flow (Alley, 1988; Azuma, 1995; Pralong et al., 2006). Accuracy of prediction is also expected to decline near the pressure melting point because additional thermally activated mechanisms, including grain boundary processes and recrystallization or enhanced mobile dislocation activity, can lead to faster than Arrhenius creep (Morgan, 1991; Cole, 2020). Moreover, microstructural and compositional influences such as salinity and brine content, porosity, and grain size and its evolution are not represented even though they can modify effective stiffness and the operative creep mechanisms in natural ice (Langleben and Pounder, 1963; Goldsby and Kohlstedt, 2001). The proposed viscosity-driven healing mechanism remains yet to be verified and uncalibrated. The triaxial behavior has been compared only at relatively high mean stress levels, on the order of 10 MPa, due to the limited availability of experimental data. These stress magnitudes are particularly relevant as they are comparable to basal overburden pressures typically encountered beneath glaciers (Herman et al., 2015, 2021; Cook et al., 2020)."*

**Rev 2:** Experimental data in Figures 3 and 4 are compared visually. Including quantitative error metrics (e.g., RMSE, R²) would significantly strengthen the model validation. We appreciate the reviewer's suggestion. Our approach follows a staged, physics-based calibration in which elastic, viscous, and damage parameters are identified from distinct regions of the response rather than from a single global least-squares fit. As a result, aggregate goodness-of-fit metrics such as $R^2$ or RMSE are not well aligned with the calibration procedure. Their values depend strongly on how the metric is defined and normalized, and can be biased by the nonuniform distribution of experimental points along

the strain axis. For these reasons, such global scores may give a misleading impression of model accuracy. It is therefore common in constitutive modeling studies to assess predictive quality primarily through direct visual comparison with experimental curves, and we have chosen to follow the same method in our study.

**Rev 2:** Table 1 presents calibrated material parameters. Please discuss any assumptions or uncertainties associated with these values and their impact on model predictions.

The following passage will be added to the revised manuscript to address this point

*"The parameters in Table 1 are calibrated for isotropic, freshwater polycrystalline ice. Under this assumption the literature indicates a comparatively narrow band for elastic constants, with Sinha (1989) reporting $E \approx 8.93$–$9.53$ GPa and $\nu \approx 0.308$–$0.311$ over $0$ to $-50\,°C$. The Maxwell configuration excludes the reversible rate-dependent deformation (primary creep), which is known to lead to lower Young's modulus values in comparison to commonly reported values. For the Glen–Arrhenius set $\{A, n, Q\}$, reported values show bounded variability with $n \approx 2$–$4$, $Q \approx 60$–$120\,\mathrm{kJ\,mol}^{-1}$, while $A$ spans orders of magnitude across studies due to microstructure and differences in calibration windows (Glen, 1955; Barnes et al., 1971; Zeitz et al., 2020). Within this context, our calibrated $\{A, n, Q\}$ lie within established intervals and are consistent with ranges reported by Durham et al. (1983) for polycrystalline ice."*

**Rev 2:** To better illustrate the influence of the damage component, please provide comparative results showing model predictions with and without the damage formulation.

The damage mechanism is essential for reproducing the post-peak brittle softening observed in the experiments. Without it, the model reduces to a simple viscoelastic Maxwell formulation, which produces only monotonic stress–strain behavior under constant strain rate. For the reviewer's reference, Fig. 1 illustrates this behavior and highlights the ne-

[Figure]

Figure 1: Comparison of calibrated model results with and without damage against triaxial compression data from Murrell et al. (1991), at $-20\,°\mathrm{C}$ with strain rate $\dot{\varepsilon}_q = 1 \times 10^{-2}\ \mathrm{s}^{-1}$.

cessity of the damage component. Because the response of such basic viscoelastic models is already well studies in the literature, we propose not to include this comparison in the revised manuscript.

**References**

Alley, R. B.: Fabrics in polar ice sheets: development and prediction, Science, 240, 493–495, https://doi.org/10.1126/science.240.4851.493, 1988.

Azuma, N.: A flow law for anisotropic polycrystalline ice under uniaxial compressive deformation, Cold reg. sci. technol., 23, 137–147, https://doi.org/10.1016/0165-232X(94)00011-L, 1995.

Barnes, P., Tabor, D., and Walker, J.: The friction and creep of polycrystalline ice, Proc. R. Soc. London, Ser. A. Mathematical and Physical Sciences, 324, 127–155, https://doi.org/10.1098/rspa.1971.0132, 1971.

Cole, D.: On the physical basis for the creep of ice: the high temperature regime, J. Glaciol., 66, 401–414, https://doi.org/10.1017/jog.2020.15, 2020.

Cook, S. J., Swift, D. A., Kirkbride, M. P., Knight, P. G., and Waller, R. I.: The empirical basis for modelling glacial erosion rates, Nat. commun., 11, 759, https://doi.org/10.1038/s41467-020-14583-8, 2020.

Durham, W., Heard, H., and Kirby, S. H.: Experimental deformation of polycrystalline H2O ice at high pressure and low temperature: Preliminary results, J. Geophys. Res.-Sol. Ea., 88, B377–B392, https://doi.org/10.1029/jb088is01p0b377, 1983.

Glen, J. W.: The creep of polycrystalline ice, P. Roy. Soc. A. Math. Phy., 228, 519–538, https://doi.org/10.1098/rspa.1955.0066, 1955.

Goldsby, D. and Kohlstedt, D. L.: Superplastic deformation of ice: Experimental observations, J. Geophys. Res.: Solid Earth, 106, 11 017–11 030, https://doi.org/10.1029/2000JB900336, 2001.

Herman, F., Beyssac, O., Brughelli, M., Lane, S. N., Leprince, S., Adatte, T., Lin, J. Y., Avouac, J.-P., and Cox, S. C.: Erosion by an Alpine glacier, Science, 350, 193–195, https://doi.org/10.1126/science.aab2386, 2015.

Herman, F., De Doncker, F., Delaney, I., Prasicek, G., and Koppes, M.: The impact of glaciers on mountain erosion, Nat. Rev. Earth Environ., 2, 422–435, https://doi.org/10.1038/s43017-021-00165-9, 2021.

Langleben, M. P. and Pounder, E.: Elastic parameters of sea ice, Massachusetts Institute of Technology, 1963.

Morgan, V.: High-temperature ice creep tests, Cold Reg. Sci. Technol., 19, 295–300, https://doi.org/10.1016/0165-232x(91)90044-h, 1991.

Murrell, S. A. F., Sammonds, P. R., and Rist, M. A.: Strength and Failure Modes of Pure Ice and Multi-Year Sea Ice under Triaxial Loading, in: Ice-Structure Interaction: IUTAM/IAHR Symposium, St. John's, Newfoundland, Canada, 1989, pp. 339–361, Springer, https://doi.org/10.1007/978-3-642-84100-2-17, 1991.

Pralong, A., Hutter, K., and Funk, M.: Anisotropic damage mechanics for viscoelastic ice, Continuum Mech. Thermodyn., 17, 387–408, https://doi.org/10.1007/s00161-005-0002-5, 2006.

Sinha, N. K.: Elasticity of natural types of polycrystalline ice, Cold Reg. Sci. Technol., 17, 127–135, https://doi.org/10.1016/s0165-232x(89)80003-5, 1989.

Zeitz, M., Levermann, A., and Winkelmann, R.: Sensitivity of ice flow to uncertainty in flow law parameters in an idealized one-dimensional geometry, TC, 2020, 1–15, https://doi.org/10.5194/tc-14-3537-2020, 2020.

---

## Author Response (AR2)

Dear Dr Roustaei,

Thank you for reviewing our revised manuscript. Following your suggestion, we have added a passage to the section *"Limitations and future validation"* that explicitly acknowledges the limitation of the Maxwell configuration in capturing primary creep. The added passage reads: *"The adopted Maxwell configuration does not capture the primary creep phase, which is commonly attributed to partially recoverable deformation and evolving internal stress fields rather than viscous flow."* An appropriate reference has also been included to support this statement. We believe this addition better reflects the nuances associated with this limitation and hope it satisfactorily addresses your comment.